# Retinal pigment epithelium drives macrophage migration during *Toxoplasma gondii* infection *in vitro*

Alex Martins Nasaré[1], Roberto Carlos Tedesco[2], Paula Andrea Faria Waziry[3], Lorena de Paula Pantaleon[4], Esther Lopes Ricci[4], Luís Antônio Baffile Leoni[5], André Rinaldi Fukushima[6]/[+], Andres Jimenez Galisteo Junior[1]

[1]Universidade de São Paulo, Faculdade de Medicina, Hospital das Clínicas, Laboratório de Investigação Médica em Protozoologia, Bacteriologia e Resistência Antimicrobiana-LIM-49, São Paulo, SP, Brasil
[2]Universidade Federal de São Paulo, São Paulo, SP, Brasil
[3]Florida Atlantic University, Schmidt College of Science, Boca Raton, FL, USA
[4]Universidade Presbiteriana Mackenzie, São Paulo, SP, Brasil
[5]Universidade Santo Amaro, São Paulo, SP, Brasil
[6]Universidade de São Paulo, Faculdade de Ciências da Saúde, São Paulo, SP, Brasil

**BACKGROUND** Ocular toxoplasmosis is a leading cause of infectious posterior uveitis worldwide. The retinal pigment epithelium (RPE), a key barrier and immunomodulatory layer in the eye, is directly targeted by *Toxoplasma gondii* during infection. However, its role in orchestrating the local immune response remains unclear.

**OBJECTIVES** To investigate whether RPE cells actively drive macrophage migration during *T. gondii* infection *in vitro*, and to identify associated cytokine profiles.

**METHODS** Adult retinal pigment epithelial cells (ARPE)-19 and primary RPE cells were exposed to tachyzoites, soluble antigens or conditioned supernatants. Macrophage migration was assessed using *Transwell®* and under-agar assays. Cytokines were quantified by cytometric bead array.

**FINDINGS** Both ARPE-19 and primary RPE exhibited chemotaxis toward parasite antigens (0.12 - 0.5 μg), and enhanced interleukin-6 (IL-6), IL-10 and tumor necrosis factor-α (TNF-α) secretion. Co-culture with RAW 264.7 macrophages further amplified cytokine production. Primary RPE from infected animals occluded 90% of *Transwell®* pores within 24h. IL-6 and IL-10 levels strongly correlated with migratory activity (r = 0.82 and 0.77, respectively).

**MAIN CONCLUSIONS** RPE cells are not passive targets but active participants in the ocular immune response to *T. gondii*. By secreting IL-6 and IL-10, they establish a chemotactic environment that recruits macrophages. These insights identify the RPE-cytokine-macrophage axis as a potential therapeutic target in ocular toxoplasmosis.

Key words: ocular toxoplasmosis - retinal pigment epithelium - macrophages - cell movement - cytokines - *Toxoplasma gondii*

The retinal pigment epithelium (RPE) is a monolayer of pigmented, highly specialized cells that absorbs excess light, recycles visual metabolites and sustains the photoreceptor layer by secreting trophic factors that preserve the choriocapillaris and the outer blood-retinal barrier.[1,2] By modulating antigen presentation and producing immunosuppressive mediators, the RPE also contributes decisively to the immune privilege of the eye.[3,4,5] Moreover, RPE cells exhibit interferon-induced antimicrobial responses.[4]

*Toxoplasma gondii* is an obligate intracellular protozoan that infects approximately one-third of the global population and is capable of invading the central nervous system and ocular tissues.[6] In the eye, the parasite most frequently induces posterior uveitis with chorioretinitis and disruption of the chorioretinal interface,[7] which may progress to necrotizing retinitis and involve the choroid, vitreous or anterior chamber.[8,9,10] Disease severity varies according to host immunity, parasite genotype and genetic background, whereas prevalence correlates with sanitation and dietary habits.[11] Human infection is usually acquired through ingestion of oocyst-contaminated water or food — particularly raw or undercooked meat — or congenitally via transplacental transmission.[6] In addition, extracellular vesicles released by *T. gondii* can trigger host immune modulation.[12]

Following ingestion, bradyzoites differentiate into tachyzoites that infect circulating monocytes, facilitating hematogenous dissemination to immune-privileged organs such as the brain and eye. Experimental evidence demonstrates that RPE cells enable the transmigration of infected monocytes across the blood-retinal

**doi:** 10.1590/0074-02760250141
**Financial support:** CAPES.
AJGJ received financial support from the FAPESP (grant 2014/26782-8). Additional financial and infrastructure support was provided by LIM-49 - Laboratório de Protozoologia, Bacteriologia e Resistência Antimicrobiana.
**+ Corresponding author:** fukushima@alumni.usp.br | ⓘ https://orcid.org/0000-0001-6026-3054

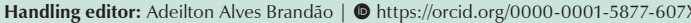

**Handling editor:** Adeilton Alves Brandão | ⓘ https://orcid.org/0000-0001-5877-607X

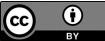

barrier, after which the parasite invades resident ocular cells; subsequent RPE migration towards infected retinal foci in *in vivo* models further disrupts retinal architecture.[10] Neutrophil-adult retinal pigment epithelial cells (ARPE-19) interactions have also been implicated in ocular toxoplasmosis.[13] Ultrastructural studies indicate that *T. gondii* manipulates host cell organelles to secure metabolic substrates[14] and modulates host-cell signaling pathways to evade immune responses,[15] indicating direct parasite-RPE communication. These findings are consistent with previous reviews describing immune subversion mechanisms employed by *T. gondii*.[16] Pro-inflammatory cytokines such as tumor necrosis factor-α (TNF-α) and interferon-γ (IFN-γ) further modulate RPE infection dynamics.[1]

Previous studies have demonstrated that ARPE-19 cells recognize and migrate towards *T. gondii*-infected cells *in vivo*,[17] suggesting an active defensive role for the retinal epithelium. The present study therefore investigates the mechanisms by which the RPE — using both the ARPE-19 cell line and primary retinal pigment epithelial (pRPE) cells — senses *T. gondii* infection and directs macrophage recruitment. The aim is to clarify how these interactions shape the pathogenesis of ocular toxoplasmosis and to identify potential therapeutic targets. Supporting this concept, recent *in vivo* transcriptomic and proteomic analyses have revealed extensive host modulation during infection.[18] Furthermore, exposure to *T. gondii* upregulates innate immunity and cytokine-related pathways in human RPE cells, particularly interleukin-6 (IL-6)/STAT3-dependent signaling,[15,19] reinforcing the concept of an active immunoregulatory role for these cells.

Based on these observations, we hypothesized that RPE cells (ARPE-19 and pRPE), when exposed to *T. gondii* antigens, actively secrete IL-6 and IL-10, thereby establishing a chemotactic gradient that recruits macrophages and modulates the local inflammatory response.

## MATERIALS AND METHODS

*ARPE-19 cell culture* - The immortalized human retinal pigment epithelial cell line ARPE-19 (ATCC CRL-2302) was kindly provided by the CASO Laboratory, Federal University of São Paulo, Brazil. Cells were maintained in Dulbecco's Modified Eagle Medium/Nutrient Mixture F-12 (DMEM/F-12; Thermo Fisher Scientific) supplemented with 10% (v/v) heat-inactivated fetal bovine serum (FBS; Gibco), 100 U mL$^{-1}$ penicillin and 100 μg mL$^{-1}$ streptomycin, at 37ºC in a humidified atmosphere containing 5% $CO_2$, as previously described.[2] Subconfluent monolayers were detached using 0.05% trypsin-EDTA and reseeded as required, including on Transwell® inserts for migration assays.

*Experimental infection of ARPE-19 cells* - ARPE-19 cultures were infected with the type II *T. gondii* ME-49 strain at a concentration of $6 \times 10^5$ tachyzoites mL$^{-1}$, corresponding to a multiplicity of infection (MOI) of 3:1. Cultures were incubated under the same conditions as uninfected controls. After 24 h, the culture medium was replaced with fresh complete DMEM/F-12. Infected cultures were passaged at least four times and are hereafter

referred to as sensitized ARPE-19 (ARPE-19-S). This experimental design was used to model immunologically primed epithelial cells and to approximate the immune activation observed in pRPE cells obtained from infected animals.

*Preparation of parasite antigen and conditioned supernatant* - The RH strain of *T. gondii* was used for antigen preparation owing to its high tachyzoite yield, whereas the cyst-forming ME-49 strain was employed for infection assays to better reproduce features of chronic infection. At 24 h post-infection, cell-free supernatants were collected, clarified by centrifugation at $3,000 \times g$ for 10 min at 4ºC, filtered through 0.22 μm membranes and stored at -20ºC until use.

*Experimental animals* - Twenty male C57BL/6J mice (eight weeks old; $20 \pm 2$ g) were obtained from the animal facility of the University of São Paulo Medical School. All experimental procedures were approved by the local ethics committee (CEUA-IMT/USP, protocol 000349A) and were conducted in accordance with the ARRIVE 2.0 guidelines.[20] Animals were acclimatized for 45 days and allocated into three experimental groups: (i) control (no intervention); (ii) immunized, receiving three intraperitoneal doses of $1 \times 10^4$ γ-irradiated ME-49 tachyzoites at 15-day intervals; and (iii) infected, receiving a single intraperitoneal dose of $1 \times 10^3$ viable ME-49 tachyzoites 30 days prior to sample collection.

*pRPE cells* - Primary mouse retinal pigment epithelial cells were isolated according to previously described protocols.[21,22] Briefly, neuroretinas were removed and eyecups were incubated in 0.2% collagenase D (Promega) diluted in DMEM for 45 min at 37ºC. Dissociated cells were cultured in complete DMEM/F-12 medium, seeded at a density of $2.5 \times 10^5$ cells per well and used up to the second passage. Culture medium was renewed every 48 h.

*ARPE-19 and RAW 264.7 co-culture* - Murine macrophages (RAW 264.7; ATCC TIB-71) were counted using a Neubauer chamber and co-cultured with ARPE-19 cells at a ratio of $1 \times 10^5$ to $5 \times 10^5$ cells (RAW:ARPE). After 24 h, five experimental conditions were established: RAW macrophages alone, ARPE-19 with RAW macrophages, sensitized ARPE-19 with RAW macrophages, ARPE-19 alone and sensitized ARPE-19 alone. Co-cultures were infected with the *T. gondii* RH strain as described above. Supernatants were collected 24 h post-infection and stored at -80ºC until analysis.

*Migration assays / Transwell® assay* - ARPE-19 or pRPE cells ($2.5 \times 10^5$ cells per insert) were seeded onto polyester Transwell® inserts with 8.0 μm pores (Corning) and cultured in complete medium for 24 h. After reaching confluence, the medium was replaced with serum-free DMEM/F-12 and the appropriate stimulus — parasite antigen, conditioned supernatant or live tachyzoites — was added to the lower chamber. After 24 h, supernatants were collected and stored at -80ºC. Cells remaining on the upper surface of the membrane were removed with cotton swabs, and membranes were fixed in 2.5% glutaraldehyde in phosphate-buffered saline (PBS) for light microscopy analysis. Cell viability was

assessed before and after each assay using Trypan Blue exclusion, with viability consistently exceeding 95%. All experiments were performed using ten independent biological replicates with technical triplicates.

*Under-agar assay* - Under-agar migration assays were performed as previously described.[23] A 1% (w/w) agar solution in PBS was autoclaved, mixed 1:1 with pre-warmed complete DMEM/F-12 and poured into six-well plates (8 mL per well). After solidification, three parallel wells (7.0 mm diameter) were created in the agar. Migration towards 10% FBS was used as a positive chemotactic control, while serum-free DMEM/F-12 served as a negative control. The central well was loaded with ARPE-19 or sensitized ARPE-19 cells ($3.8 \times 10^2$ cells mm$^{-2}$), while lateral wells contained either control media, serial dilutions of RH antigen (0.12 - 2.0 µg), conditioned supernatant or viable tachyzoites. After 24 h, supernatants were collected for cytokine analysis. Cells were then fixed in 4% paraformaldehyde, stained with DAPI and imaged by fluorescence microscopy. Migration distances were quantified using ImageJ (FIJI version).

*Microscopy / light microscopy* - Fixed Transwell® membranes were dehydrated in graded ethanol solutions, infiltrated overnight with a mixture of Technovit 7100 (glycol methacrylate) and ethanol (1:1), embedded in resin, sectioned at 1 µm thickness and stained with 0.25% toluidine blue in 1% sodium borate. Sections were mounted using Erv-Mont™ mounting medium and examined under a Nikon Eclipse light microscope.

*Transmission electron microscopy* - Confluent pRPE monolayers grown on 0.8 µm-pore inserts were serum-starved for 24 h and exposed to $1.2 \times 10^5$ ME-49 tachyzoites added to the upper compartment without direct cell contact. Pre- and post-exposure supernatants were collected and stored at -80ºC. Cells were fixed in 2.5% glutaraldehyde, post-fixed in 1% osmium tetroxide, dehydrated, embedded in epoxy resin, ultrathin-sectioned and examined using a JEOL JEM-1010 transmission electron microscope. Environmental factors influencing cytokine signaling in retinal pigment epithelial cells were considered during experimental design.[24]

*Cytokine quantification* - Supernatants were diluted 1:1 in BD FACSFlow™ sheath fluid and analyzed using a BD LSRFortessa flow cytometer with the BD Cytometric Bead Array Mouse Cytokine Kit, following the manufacturer's instructions. Data acquisition was performed using FACSDiva software version 8.0 and analyzed with FlowJo version 10.8.

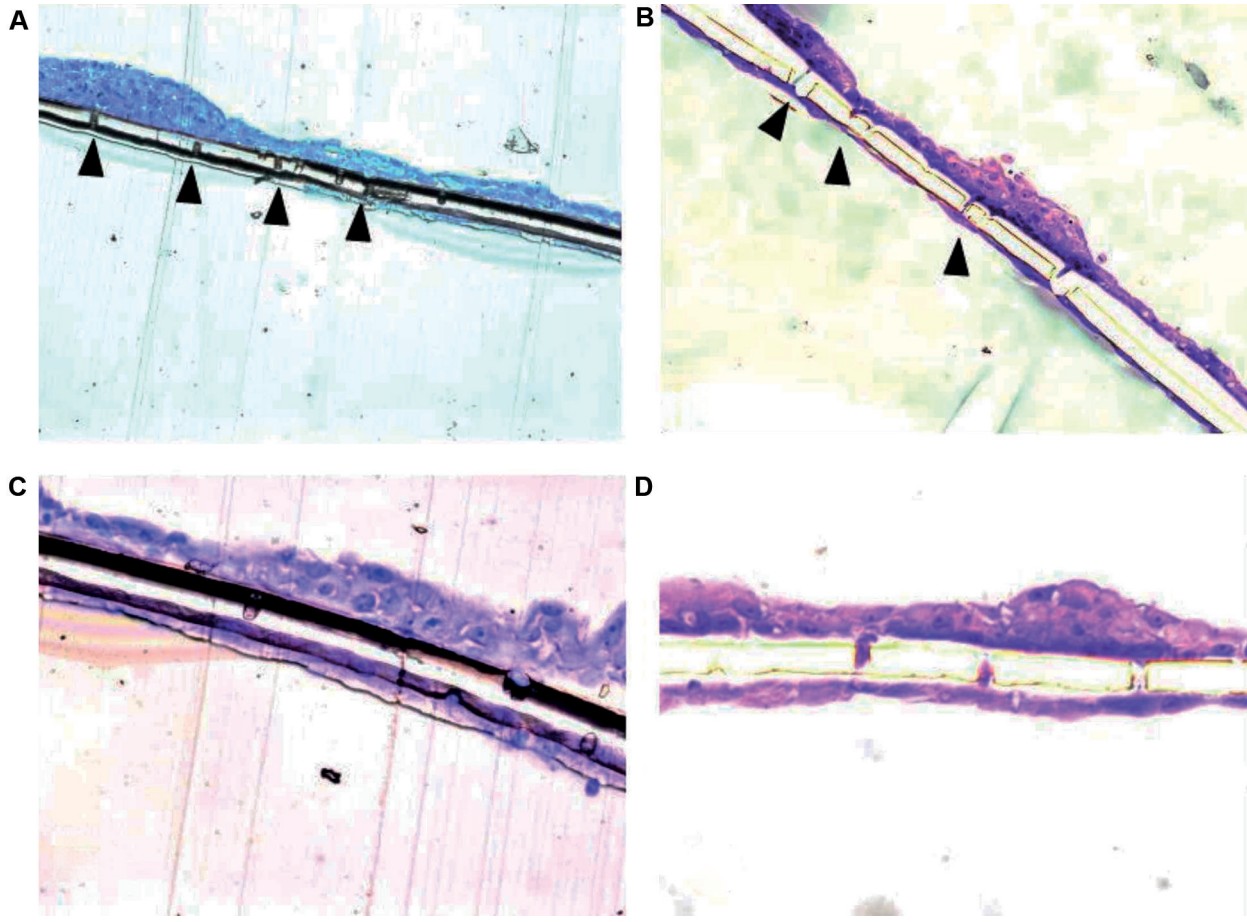

Fig. 1: transmembrane migration of adult retinal pigment epithelial (ARPE-19) cells through Transwell® insert pores to the lower surface of the membrane. (A, B) ARPE-19 cells showing gaps (arrows) corresponding to pores on the lower membrane surface without cell coverage. (C, D) Sensitised ARPE-19 cells (ARPE-19-S) forming a continuous monolayer completely covering the pores.

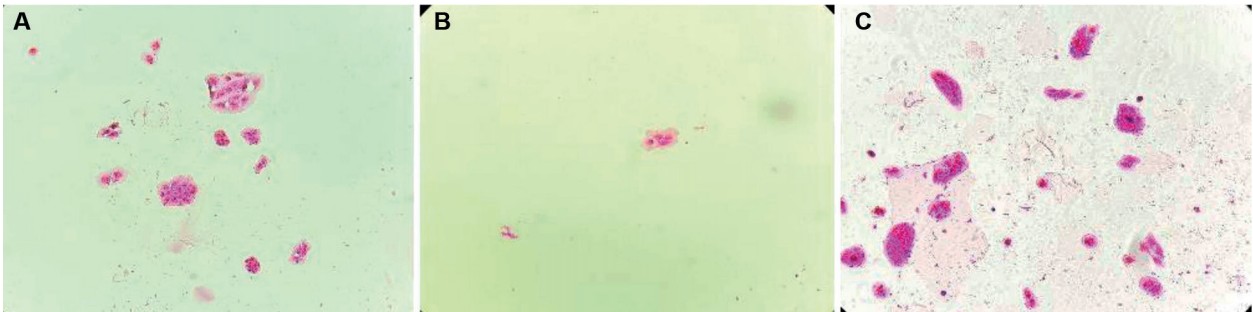

Fig. 2: Transwell® migration assay showing adult retinal pigment epithelial (ARPE-19) cell migration through membrane pores and adhesion to the lower surface of the well. (A) Control condition. (B) Exposure to *Toxoplasma gondii* RH strain ($2.5 \times 10^{-3}$ µg/µL). (C) Exposure to soluble *T. gondii* antigen. Both stimuli increased the migratory response.

Fig. 3: chemotaxis of adult retinal pigment epithelial (ARPE-19) and sensitised ARPE-19 cells in the under-agar assay after 24 h. Representative images under different experimental conditions. Arrows indicate the migration front.

*Statistical analysis* - Statistical analyses were performed using GraphPad Prism version 9.0 (GraphPad Software, San Diego, CA, USA). Group means were compared using one-way analysis of variance (ANOVA) followed by Tukey's post-hoc test. Values of $p < 0.05$ were considered statistically significant. Effect sizes were calculated using Cohen's d and interpreted as low ($d < 0.5$), medium ($0.5 \leq d < 0.8$) or high ($d \geq 0.8$). All raw data, processed images and analysis scripts are available from the corresponding author upon reasonable request.

## RESULTS

*ARPE-19 migration in Transwell® assays* - Both non-infected ARPE-19 and sensitized ARPE-19 (ARPE-19-S) monolayers were able to traverse the 8.0 µm-pore Transwell® membranes. On the lower surface of the

membranes, ARPE-19-S cells formed a continuous layer completely covering the pores, whereas ARPE-19 cells displayed focal gaps corresponding to pores devoid of cells (Fig. 1A-D). Quantitative analysis of pore occupancy confirmed a significantly higher proportion of fully occluded pores in ARPE-19-S cultures compared with ARPE-19 cultures (p < 0.05). Migrated ARPE-19 cells were also observed adhering to the lower surface of the insert membrane and, in some cases, detaching and settling at the bottom of the well (Fig. 2A-C).

*Chemotaxis measured by the under-agar assay* - ARPE-19 cell migration in the under-agar assay exhibited a clear dose-dependent pattern. Migration distances were greatest at lower concentrations of soluble *T. gondii* antigen (0.12-0.50 µg), with median distances exceeding 500 µm after 24 h. At higher antigen concentrations (1.0-2.0 µg), migration was markedly reduced and approached values observed under negative control conditions (Figs 3A-L, 4A). Neither conditioned supernatants nor live tachyzoites induced measurable directional migration.

Fig. 4: quantification of under-agar migration. (A) Median migration distance after 24 h. (B) Number of cells crossing the agar interface. (C) Comparison of migration at 24 h and 48 h. Values are mean ± standard deviation (SD) (n = 5).

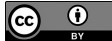

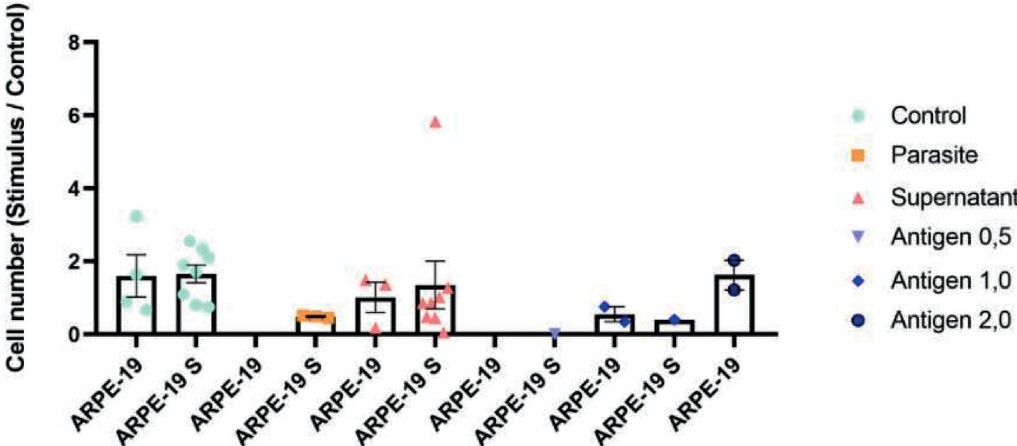

Fig. 5: quantification of adult retinal pigment epithelial (ARPE-19) cell migration in the Transwell® assay after exposure to different stimuli. Bars represent the number of cells in the lower chamber following exposure to live tachyzoites, conditioned supernatant or soluble antigen (0.12-2.0 μg). Values are mean ± standard deviation (SD) (n = 3). *p < 0.05 versus control [analysis of variance (ANOVA)/Tukey].

The total number of cells crossing the agar interface did not differ significantly among experimental groups (Figs 4B, 5). Prolongation of the incubation period to 72 h did not alter the observed migration profile (Fig. 4C).

*Cytokine secretion by ARPE-19 cultures* - Supernatants collected from Transwell® migration assays contained detectable levels of IL-6, IL-10 and TNF-α. Peak concentrations of these cytokines coincided with the antigen concentrations (0.12-0.50 μg) that induced maximal migratory responses. Strong positive correlations were observed between migration distance and cytokine levels for IL-6 (r = 0.82, p < 0.01), IL-10 (r = 0.77, p < 0.05) and TNF-α (r = 0.74, p < 0.05; Pearson correlation). IL-2 was detected predominantly under non-migratory conditions and was absent in sensitized ARPE-19 cultures exposed to live tachyzoites, suggesting that IL-2 release was not associated with chemotactic activity (Table I).

*Cytokine profile of ARPE-19-RAW 264.7 co-cultures* - Exposure of ARPE-19 cells to *T. gondii* tachyzoites resulted in approximately twofold increases in IL-6 and IL-10 secretion. Co-culture with RAW 264.7 macrophages markedly amplified cytokine production, with IL-6 concentrations reaching up to 362 pg mL⁻¹ and IL-10 levels up to 343 pg mL⁻¹. Infected co-cultures exhibited even higher cytokine levels, producing up to 842 pg mL⁻¹ IL-6 and 471 pg mL⁻¹ IL-10, along with substantial TNF-α secretion (742 pg mL⁻¹). Effect size analysis using Cohen's d classified the increases in IL-6, IL-10 and TNF-α as high across parasite-stimulated co-culture conditions (Table II).

*Migration of pRPE cells* - pRPE cells isolated from control mice exhibited limited migratory activity and preserved plasma membrane integrity, with partial pore occupancy observed after 24 h (Fig. 6A). Cells derived from immunized animals occupied a larger proportion of Transwell® membrane pores and displayed numerous cytoplasmic vesicles and membrane protrusions (Fig. 6B-C). In contrast, pRPE cells from infected animals showed extensive membrane disruption and virtually complete pore occlusion, indicating intense migratory activity (Fig. 6D-F).

*Cytokine secretion by pRPE cells* - Basal pRPE cultures secreted high levels of IL-6 (788 pg mL⁻¹), low levels of TNF-α (5.9 pg mL⁻¹) and no detectable IL-10. pRPE cells from immunized animals produced lower IL-6 concentrations (414 pg mL⁻¹) while maintaining similar TNF-α levels. Infected pRPE cultures secreted markedly reduced IL-6 levels (153 pg mL⁻¹) but exhibited the highest IL-10 concentrations (6.6 pg mL⁻¹). TNF-α concentrations remained relatively constant across all experimental groups (approximately 6 pg mL⁻¹) (Fig. 7A-C).

## DISCUSSION

Ocular toxoplasmosis remains one of the leading infectious causes of posterior uveitis and vision loss worldwide, yet the cellular events that initiate retinal damage remain poorly defined. The present study demonstrates that human and murine RPE actively detects and responds to *T. gondii*, orchestrating a chemotactic axis dominated by IL-6 and IL-10 *in vitro*. Three key findings emerge from our results.

*RPE is intrinsically migratory and dose-responsive* - ARPE-19 and pRPE cells migrated through Transwell® membranes and under-agar gradients, but only when exposed to soluble *T. gondii* antigen concentrations ≤ 0.5 μg. Higher doses abolished chemotaxis, indicating a bell-shaped dose-response curve similar to that described for professional phagocytes.[13] These data extend previous *in vivo* findings showing that RPE cells migrate toward intraretinal parasites[17,25] and confirm that this response can be reliably reproduced *in vitro*. Comparable IL-6/IL-10-driven communication between epithelial and macrophage compartments has been observed in intestinal and pulmonary tissues,[26] suggesting that RPE cells share conserved immunomodulatory mechanisms with other mucosal barriers. Similar transcriptional changes in human RPE during *T. gondii* infection have also been documented.[18]

Fig. 6: migration of primary retinal pigment epithelial (pRPE) cells analysed by transmission electron microscopy. (A, B) Cells from control animals. (C, D) Cells from immunised animals showing membrane protrusions traversing the pores (arrows). (E, F) Cells from infected animals showing complete pore occlusion and disrupted membranes.

*IL-6, IL-10 and TNF-α correlate with migration intensity* - Soluble antigen concentrations that maximized chemotaxis also induced the greatest secretion of IL-6, IL-10 and TNF-α, whereas IL-2 was associated with non-migratory conditions.[22] Co-culture with RAW 264.7 macrophages amplified IL-6, IL-10 and TNF-α release, reinforcing the dynamic communication between retinal epithelial cells and macrophages.[27,28,29] IL-6 may act as an autocrine primer that upregulates IL-1 expression, providing negative feedback to limit

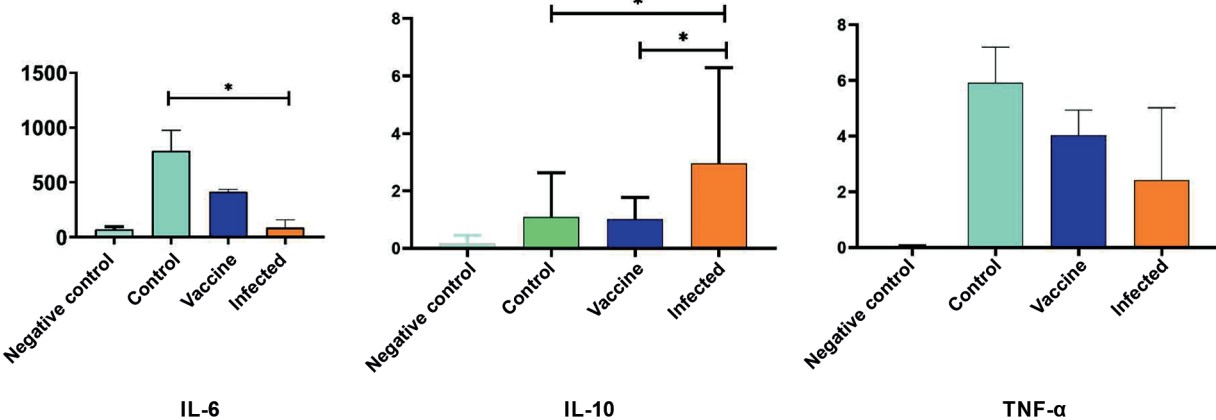

Fig. 7: cytokine secretion by primary retinal pigment epithelial (pRPE) cells after exposure to *Toxoplasma gondii* ME-49 strain. (A) interleukin (IL)-6. (B) IL-10. (C) tumour necrosis factor-alpha (TNF-α). Groups: basal, control, immunised and infected. Values are expressed as pg mL⁻¹ [mean ± standard deviation (SD), n = 3]. *p < 0.05 [analysis of variance (ANOVA)/Tukey].

TABLE I

Cytokines secreted by adult retinal pigment epithelial (ARPE-19) and sensitised ARPE-19 cells (ARPE-19-S) after stimulation. Relative expression (z-scores) of interleukin-6 (IL-6), IL-10, tumour necrosis factor-alpha (TNF-α) and IL-2 following exposure to live *Toxoplasma gondii* tachyzoites, conditioned supernatant or soluble antigen (0.12-2.0 µg). Green indicates values approaching 1, and red indicates values approaching -1

|  |  | Parasite | Supernatant | 0,5 µg | 1 µg | 2 µg | 0,25 µg | 0,125 µg |
|---|---|---|---|---|---|---|---|---|
| IL-6 | ARPE-19 | 0,8164 | 0,8164 | 0,8164 | 0,8164 | 0,6237 | 0,8164 | 0,8164 |
|  | ARPE-19 S | 0,8165 | 0,8165 | 0,8165 | 0,8165 | 0,8165 | -1,1466 | 0,8165 |
| IL-10 | ARPE-19 | 0,8793 | 0,8793 | 0,8793 | 0,8793 | 0,8793 | 0,7049 | 0,6762 |
|  | ARPE-19 S | 0 | 0 | 0 | 0 | 0 | 0 | -0,8164 |
| TNF-α | ARPE-19 | 0,8164 | 0,8164 | 0,8164 | 0,3495 | 0,8164 | 0,8164 | 0,8164 |
|  | ARPE-19 S | 0 | 0 | 0 | 0 | 0 | -0,8164 | 0 |
| IL-2 | ARPE-19 | 0,8391 | 0,8672 | 0,8774 | 1,2624 | 0,8112 | 0,5896 | 0,8515 |
|  | ARPE-19 S | 1,2838 | 1,2838 | 1,2838 | 1,035 | 0,9979 | 0,6928 | 0,9137 |

| 1 | 0,8 | 0,6 | 0,4 | 0,2 | 0 | -0,2 | -1 |
|---|---|---|---|---|---|---|---|

tissue damage.[2,16] The moderate TNF-α concentrations detected are consistent with a local, protective response that restrains parasite replication without compromising RPE integrity.[19,21]

*Prior exposure to T. gondii primes pRPE motility* - pRPE cells from immunized or chronically infected mice occluded nearly every Transwell® pore within 24 h, whereas control cells required at least 48 h. This priming was accompanied by elevated IL-10 levels in infected animals and sustained IL-6 production in control cells, as previously described in experimental models of *T. gondii* infection,[21] suggesting that systemic antigen exposure conditions the ocular microenvironment,[5] thereby accelerating pRPE chemotaxis during subsequent challenges.[15] These cytokine profiles closely resemble those reported in recurrent toxoplasmic retinochoroiditis.[30]

*Proposed model* - Upon retinal exposure to low concentrations of *T. gondii* antigen, RPE cells secrete IL-6 and TNF-α, initiating a controlled pro-inflammatory milieu that attracts macrophages and promotes their own migration toward the infectious focus. As IL-10 levels rise, this cytokine attenuates IL-6 signaling, preserving retinal architecture yet inadvertently favoring parasite persistence, a balance that may underlie recurrent clinical relapses.[14] Conversely, excessive antigen exposure or an overwhelming tachyzoite load disrupts this regulatory axis, suppressing chemotaxis and potentially permitting uncontrolled parasite dissemination.

*Study limitations and future directions* - The present study relied exclusively on *in vitro* assays, which may not fully recapitulate the *in vivo* retinal microenvironment. Validation using retinal organoids or animal models will therefore be crucial to confirm the physiological relevance of the IL-6/IL-10 axis. The ARPE-19 cell line, although widely used,[7,26] does not fully reproduce the polarity or phagocytic capacity of primary RPE cells.

TABLE II

Cytokine concentrations and effect sizes in co-cultures of adult retinal pigment epithelial (ARPE-19) cells and RAW 264.7 murine macrophages. Levels of interleukin-6 (IL-6), IL-4, IL-10, IL-2 and tumour necrosis factor-alpha (TNF-α) (pg mL⁻¹) measured 24 h after exposure to *Toxoplasma gondii* RH strain. Effect sizes were calculated using Cohen's d and classified as null, low, medium or high. Values represent the meaning of three independent experiments

| | | Control | | Parasite | | Size effect | D de Cohen |
|---|---|---|---|---|---|---|---|
| IL-6 | RAW | 0 | 0 | 0 | 0 | 0 | Null |
| | RAW + ARPE - 19 | 0 | 0,41 | 362,13 | 0,7 | 1 | High |
| | RAW + ARPE - 19 S | 5,46 | 3,31 | 841,97 | 1983,91 | 2 | High |
| | ARPE - 19 | 0 | 1,88 | 0,6 | 0 | 0,6 | Medium |
| | ARPE - 19 S | 0 | 0,11 | 2,28 | 0 | 0,9 | High |
| IL-4 | RAW | 0 | 0 | 0 | 0 | 0 | Null |
| | RAW + ARPE - 19 | 0,53 | 1,4 | 0,95 | 1,29 | 0,33 | Low |
| | RAW + ARPE - 19 S | 0,35 | 0 | 0 | 1,15 | 0,66 | Medium |
| | ARPE - 19 | 0,3 | 3,42 | 0 | 0 | 1,19 | High |
| | ARPE - 19 S | 0 | 0,39 | 0,76 | 0 | 0,4 | Low |
| IL-10 | RAW | 0 | 0 | 0 | 0 | 0 | Null |
| | RAW + ARPE - 19 | 1,61 | 0 | 343,35 | 247,53 | 6,14 | High |
| | RAW + ARPE - 19 S | 12,45 | 3,81 | 471,43 | 332,94 | 5,67 | High |
| | ARPE - 19 | 0 | 17,74 | 1,61 | 0 | 0,9 | High |
| | ARPE - 19 S | 0 | 0 | 4,05 | 0 | 1 | High |
| IL-2 | RAW | 0 | 0 | 0 | 0 | 0 | Null |
| | RAW + ARPE - 19 | 1,02 | 0,74 | 1,33 | 0,87 | 0,8 | High |
| | RAW + ARPE - 19 S | 0 | 1,08 | 1,38 | 0,72 | 0,8 | High |
| | ARPE - 19 | 1,08 | 2,38 | 1,29 | 0,51 | 1,09 | High |
| | ARPE - 19 S | 0 | 1,73 | 0 | 0 | 1 | High |
| TNF-α | RAW | 0 | 0 | 0 | 0 | 0 | Null |
| | RAW + ARPE - 19 | 270,34 | 350,5 | 400,2 | 27,78 | 0,5 | Medium |
| | RAW + ARPE - 19 S | 2423,33 | 2290,84 | 742,25 | 1152,09 | 6,5 | High |
| | ARPE - 19 | 0 | 11,96 | 0 | 0 | 1 | High |
| | ARPE - 19 S | 5,79 | 0 | 1,88 | 0 | 0,64 | Medium |

| 0 | 1 | 2 | 3 | 4 | 5 | 6 | 7 | 8 | 9 | 10 |

All numerical data are presented as mean ± standard deviation (SD) (n = 5). Statistical comparisons were performed by one-way analysis of variance (ANOVA) followed by Tukey's post-hoc test; *$p < 0.05$.

[31] Only early-passage primary cells were analyzed to minimize culture artefacts; however, limited yields constrained the performance of more detailed mechanistic assays. Future investigations should employ organoid or explant models to verify cytokine gradients *in situ*, as previously explored in experimental infection models,[27] dissect downstream signaling pathways — particularly STAT3- and STAT6-dependent signaling — and assess whether pharmacological modulation of this axis mitigates experimental ocular toxoplasmosis.

*Clinical implications* - These findings open new avenues for the therapeutic modulation of cytokine balance in ocular toxoplasmosis, offering opportunities for integrated immunomodulatory and antiparasitic strategies. Targeting the IL-6/IL-10 axis may provide a dual advan-

tage by enhancing parasite clearance while minimizing collateral retinal injury. Agents that transiently enhance IL-6 signaling or antagonize IL-10 at early stages might strengthen host defenses, whereas sustained IL-6 inhibition could prevent chronic inflammation once parasite burden is controlled. Such stage-specific interventions warrant evaluation in preclinical models.

*Concluding remarks* - This study demonstrates, for the first time, that RPE cells exhibit a dose-dependent chemotactic response to *T. gondii* antigens, governed by the coordinated action of IL-6 and IL-10. These findings establish a mechanistic link between epithelial activation and macrophage recruitment in ocular toxoplasmosis, identifying a potential therapeutic target for the modulation of retinal inflammation. This migrato-

ry behavior is accompanied by a cytokine profile dominated by IL-6 and TNF-α, subsequently attenuated by IL-10. Collectively, the results support a model in which ARPE-19 cells and pRPE cells detect the parasite, initiate a controlled pro-inflammatory response to recruit immune cells, and later limit tissue damage through IL-10-mediated feedback. Targeted modulation of the IL-6/TNF-α/IL-10 axis therefore emerges as a promising strategy to mitigate retinal injury while preserving host defenses in ocular toxoplasmosis. The authors declare no competing interests.

## ACKNOWLEDGEMENTS

To Paula Andrea Faria Waziry for technical and language support and for assistance with data analysis. The authors would like to express their sincere gratitude to Dr Cleusa Fumica Hirata Takakura, technician of the Microscopy Laboratory at the Faculdade de Medicina da Universidade de São Paulo (FMUSP), for her valuable assistance in the acquisition of transmission electron microscopy images.

## AUTHORS' CONTRIBUTION

AMN conceived and designed the study, performed the experiments, analysed the data and drafted the manuscript; RCT contributed to methodology and histological analysis and reviewed the manuscript; PAFW contributed to study conception and manuscript review; LPP performed cell migration assays, curated the data and contributed to data visualization; ELR performed immunofluorescence imaging and contributed to data curation and manuscript review; LABL and ARF supervised the study and reviewed the manuscript; AJGJ provided resources, performed the statistical analysis, acquired funding and reviewed the manuscript.

## DATA AVAILABILITY

All data generated and analysed during this study are included in the manuscript or are available from the corresponding author upon reasonable request.

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

# OPEN PEER REVIEW

Memórias do IOC thanks the anonymous reviewers for their contribution to the peer review of this work.

**FIRST REVIEW ROUND**

REVIEWERS' COMMENTS

**REVIEWER #1**

General comments

Throughout the manuscript, italics should be consistently used for Toxoplasma gondii and other Latin expressions (e.g., in vitro).

"In the Materials and Methods section, the authors refer to the ARPE-19 cell line, but in the Introduction and other parts of the text they use the term 'RPE cells.' Please ensure consistency throughout the manuscript. If the authors intend to refer to primary RPE (pRPE) cells, this should be clarified."

The introduction is clear for the paper, linking RPE and immunity with T.gondii pathogenesis. It would be helpful to clarify if the RPE migration toward infected foci is based on in vivo or in vitro, or both (Line 42).

In several figures, the panel letters appear incomplete or difficult to read. Please revise and replace the labels to ensure they are clear and consistent, which will improve figure interpretation.

The section is generally well described, but it is unclear why the RH strain was used for the antigen preparation while ME49 was used for culture infection. In the co-culture assays, it would be helpful to provide details on cell densities and how cell viability was controlled. In addition, I recommend clarifying the number of experimental replicates in the Statistical Analysis section.

Results:

The findings are relevant and interesting but poorly described.

The description of Figure 1 should specify where the cells migrated, under what conditions, and provide appropriate comparisons. More details are needed. If the authors only state that 'migration' occurred, it could be erroneously interpreted that the cells migrated to the bottom. Did the cells migrate to the bottom of the Transwell insert, or to another location? This information should be incorporated into the main text rather than left only in the figure legend.

I recommend avoiding the expression 'cell displacement,' as it may lead to misinterpretation (for example, section 3.2 Chemotaxis … ). Both tachyzoites and ARPE cells are cells, so the term is ambiguous. Given that the results are not clearly described, this type of wording makes it particularly difficult to follow the interpretation.

Some of the figures are missing figure numbers, which makes the revision process difficult. For example, the figure titled 'Migration of primary retinal pigment epithelium' should be properly numbered to ensure consistency and clarity throughout the manuscript.

Figure 1 (Section 3.1): Please correct the letter 'B,' which currently looks like an 'E.' In addition, specify how pore occlusion was quantified and indicate the number of replicates performed.

Figure 2 (Section 3.1): Please also correct the figure panel letters to ensure clarity.

Figure 3 (Section 3.2, under-agar assay): If possible, I suggest repeating the control conditions for ARPE-19, ARPE-19S, ARPE-19 (antigen 0.5), and ARPE-19 (antigen 1.0). I also recommend maintaining the same scale when comparing migration differences at 24 h vs. 48 h for consistency and clarity."

In section 3.3, the detection of IL-2 in non-migratory conditions should be explained in more detail, and I don't understand why the values of exposure to live tachyzoites and conditioned supernatant are the same, Can you clarify that?

For the co-culture experiments (section 3.4), it would be helpful to specify whether RAW cells infected alone were included as controls. Regarding the RPE experiments (sections 3.5–3.6), the data are interesting, but please double-check the citation of Figure 4, as it seems incorrect I believe there are six figures in total, not five

In the Discussion, I would suggest moderating statements such as 'RPE orchestrates a chemotactic axis', since the current evidence is based on in vitro assays and may not fully capture the in vivo complexity. Similarly, the statement that 'systemic antigen encounter conditions the ocular microenvironment' should be nuanced, as only isolated pRPE were studied. In the Conclusion, I recommend using 'our findings suggest' and explicitly acknowledging the need for validation in organoid or animal models before clinical translation.

In section 3.3, the detection of IL-2 under non-migratory conditions should be explained in more detail. It is also unclear why the values obtained for exposure to live tachyzoites and conditioned supernatant appear to be the same—could the authors clarify this?

For the co-culture experiments (section 3.4), please specify whether RAW cells infected alone were included as controls.

Regarding the RPE experiments (sections 3.5–3.6), the data are interesting, but please double-check the citation of Figure 4, as it seems incorrect. I believe there are six figures in total, not five.

Discussion

I suggest moderating statements such as "RPE orchestrates a chemotactic axis," since the evidence provided is based solely on in vitro assays and may not fully reflect the in vivo context.

Similarly, the claim that "systemic antigen encounter conditions the ocular microenvironment" should be nuanced, as only isolated pRPE were studied.

I recommend rephrasing to use wording such as "our findings suggest," and explicitly acknowledging the need for validation in organoid or animal models before any clinical translation.

## AUTHORS' RESPONSE TO THE REVIEWERS

"In the Materials and Methods section, the authors refer to the ARPE-19 cell line, but in the Introduction and other parts of the text they use the term 'RPE cells.' Please ensure consistency throughout the manuscript. If the authors intend to refer to primary RPE (pRPE) cells, this should be clarified."

Cell types were clarified and consistently identified as either *pRPE* or *ARPE-19* throughout the manuscript.

The introduction is clear for the paper, linking RPE and immunity with T.gondii pathogenesis. It would be helpful to clarify if the RPE migration toward infected foci is based on in vivo or in vitro, or both (Line 42).

A paragraph was added clarifying that RPE migration was evaluated under *in vitro* conditions.

In several figures, the panel letters appear incomplete or difficult to read. Please revise and replace the labels to ensure they are clear and consistent, which will improve figure interpretation.

All figures were revised, and panel labels were replaced to ensure clarity and consistency

The section is generally well described, but it is unclear why the RH strain was used for the antigen preparation while ME49 was used for culture infection. In the co-culture assays, it would be helpful to provide details on cell densities and how cell viability was controlled. In addition, I recommend clarifying the number of experimental replicates in the Statistical Analysis section.

A paragraph was added explaining the rationale for using RH and ME49 strains.

The number of experimental replicates was specified in the Statistical Analysis section.

Results:

The findings are relevant and interesting but poorly described.

The description of Figure 1 should specify where the cells migrated, under what conditions, and provide appropriate comparisons. More details are needed. If the authors only state that 'migration' occurred, it could be erroneously interpreted that the cells migrated to the bottom. Did the cells migrate to the bottom of the Transwell insert, or to another location? This information should be incorporated into the main text rather than left only in the figure legend.

The paragraph describing Figure 1 was rewritten to include details on migration sites, conditions, and comparisons.

I recommend avoiding the expression 'cell displacement,' as it may lead to misinterpretation (for example, section 3.2 Chemotaxis … ). Both tachyzoites and ARPE cells are cells, so the term is ambiguous. Given that the results are not clearly described, this type of wording makes it particularly difficult to follow the interpretation.

Terminology was revised to avoid ambiguity and improve precision.

Some of the figures are missing figure numbers, which makes the revision process difficult. For example, the figure titled 'Migration of primary retinal pigment epithelium' should be properly numbered to ensure consistency and clarity throughout the manuscript.

Figure 1 (Section 3.1): Please correct the letter 'B,' which currently looks like an 'E.' In addition, specify how pore occlusion was quantified and indicate the number of replicates performed.

Figure numbers were corrected and standardized throughout the manuscript.

The letter was corrected. Quantification methods and the number of replicates were included in the text.

Figure 2 (Section 3.1): Please also correct the figure panel letters to ensure clarity.

Figure 2 panel letters were corrected for consistency and clarity.

Figure 3 (Section 3.2, under-agar assay): If possible, I suggest repeating the control conditions for ARPE-19, ARPE-19S, ARPE-19 (antigen 0.5), and ARPE-19 (antigen 1.0). I also recommend maintaining the same scale when comparing migration differences at 24 h vs. 48 h for consistency and clarity."

Triplicate numbers and experimental conditions were indicated for each situation. The same scale was maintained across comparisons for consistency.

In section 3.3, the detection of IL-2 in non-migratory conditions should be explainded in more detail, and I don't understand why the values of exposure to live tachyzoites and conditioned supernatant are the same, Can you clarify that?

A paragraph was added explaining IL-2 detection results under non-migratory conditions.

For the co-culture experiments (section 3.4), it would be helpful to specify whether RAW cells infected alone were included as controls. Regarding the RPE experiments (sections 3.5–3.6), the data are interesting, but please double-check the citation of Figure 4, as it seems incorrect I believe there are six figures in total, not five

Figure numbering was verified and corrected throughout the manuscript.

In the Discussion, I would suggest moderating statements such as 'RPE orchestrates a chemotactic axis', since the current evidence is based on in vitro assays and may not fully capture the in vivo complexity. Similarly, the statement that 'systemic antigen encounter conditions the ocular microenvironment' should be nuanced, as only isolated pRPE were studied.

Statements were revised to moderate the conclusions and acknowledge the *in vitro* limitations.

In the Conclusion, I recommend using 'our findings suggest' and explicitly acknowledging the need for validation in organoid or animal models before clinical translation.

The suggested phrasing was adopted, and a note on future validation in organoid and animal models was included.

## SECOND REVIEW ROUND

**REVIEWERS' COMMENTS**

### REVIEWER #1

The authors have addressed my previous comments, including revising figures, correcting numbering, and adding data to improve the clarity of the manuscript description. They have also added sentences to moderate their conclusions and acknowledge the limitations of the in vitro model. I would only like to suggest three minor points:

1) In my previous comment, I mentioned:

"The description of Figure 1 should specify where the cells migrated, under what conditions, and provide appropriate comparisons. More details are needed. If the authors only state that 'migration' occurred, it could be erroneously interpreted that the cells migrated to the bottom. Did the cells migrate to the bottom of the Transwell insert or to another location? This information should be incorporated into the main text rather than left only in the figure legend."

However, the authors did not present any corresponding results; they only described a result without showing supporting data. Data not shown should either be presented in the main text or included as supplemental material.

2) It will also benefit the manuscript if the authors clarify why and how they decided to use a sensitized cell line.

3) Finally, please standardize the term "primary RPE" as "pRPE" throughout the text.

**AUTHORS' RESPONSE TO THE REVIEWERS**

Response to the Editor

As requested, a justification regarding the ARPE-19 cell line was added at the end of section 2.2. Experimental infection of ARPE-19 cells.

The term Primary RPE was standardized as pRPE throughout the manuscript.

A new result describing the final destination of the experimental cells—previously referred to as *data not shown*—was included (Figure 2A–C) and is now explained in section 3.1. ARPE-19 migration in Transwell assays. Accordingly, the legend of Figure 1 was revised, and Figure 2 was added. All figures were renumbered, and the legend of Figure 4 was rewritten for clarity.

## THIRD REVIEW ROUND

**REVIEWERS' COMMENTS**

### REVIEWER #1

The authors have adequately addressed all the aspects previously suggested. The abstract is clear and accurately reflects the scope and findings of the work. The contribution remains original and relevant, representing an important addition to the field. The methodology, results, and discussion are coherent and well-structured, providing sufficient detail to support the conclusions. Results in general are clearly presented, informative, and consistent with the text.

