## [Reviewer Report · FIRST REVIEW ROUND - REVIEWERS COMMENTS]

## REVIEWER #1

**General comments**

Throughout the manuscript, italics should be consistently used for *Toxoplasma gondii* and other Latin expressions (e.g., *in vitro*).

“In the Materials and Methods section, the authors refer to the ARPE-19 cell line, but in the Introduction and other parts of the text they use the term ‘RPE cells.’ Please ensure consistency throughout the manuscript. If the authors intend to refer to primary RPE (pRPE) cells, this should be clarified.”

The introduction is clear for the paper, linking RPE and immunity with *T.gondii* pathogenesis.

It would be helpful to clarify if the RPE migration toward infected foci is based on *in vivo* or *in vitro*, or both (Line 42).

In several figures, the panel letters appear incomplete or difficult to read.

Please revise and replace the labels to ensure they are clear and consistent, which will improve figure interpretation.

The section is generally well described, but it is unclear why the RH strain was used for the antigen preparation while ME49 was used for culture infection.

In the co-culture assays, it would be helpful to provide details on cell densities and how cell viability was controlled.

In addition, I recommend clarifying the number of experimental replicates in the Statistical Analysis section.

**Results:**

The findings are relevant and interesting but poorly described.

The description of Figure 1 should specify where the cells migrated, under what conditions, and provide appropriate comparisons.

More details are needed. If the authors only state that ‘migration’ occurred, it could be erroneously interpreted that the cells migrated to the bottom.

Did the cells migrate to the bottom of the Transwell insert, or to another location?

This information should be incorporated into the main text rather than left only in the figure legend.

I recommend avoiding the expression ‘cell displacement,’ as it may lead to misinterpretation (for example, section 3.2 Chemotaxis … ).

Both tachyzoites and ARPE cells are cells, so the term is ambiguous.

Given that the results are not clearly described, this type of wording makes it particularly difficult to follow the interpretation.

Some of the figures are missing figure numbers, which makes the revision process difficult.

For example, the figure titled ‘Migration of primary retinal pigment epithelium’ should be properly numbered to ensure consistency and clarity throughout the manuscript.

Figure 1 (Section 3.1): Please correct the letter ‘B,’ which currently looks like an ‘E.’ In addition, specify how pore occlusion was quantified and indicate the number of replicates performed.

Figure 2 (Section 3.1): Please also correct the figure panel letters to ensure clarity.

Figure 3 (Section 3.2, under-agar assay): If possible, I suggest repeating the control conditions for ARPE-19, ARPE-19S, ARPE-19 (antigen 0.5), and ARPE-19 (antigen 1.0).

I also recommend maintaining the same scale when comparing migration differences at 24 h vs. 48 h for consistency and clarity.”

In section 3.3, the detection of IL-2 in non-migratory conditions should be explainded in more detail, and I don’t understand why the values of exposure to live tachyzoites and conditioned supernatant are the same, Can you clarify that?

For the co-culture experiments (section 3.4), it would be helpful to specify whether RAW cells infected alone were included as controls.

Regarding the RPE experiments (sections 3.5–3.6), the data are interesting, but please double-check the citation of Figure 4, as it seems incorrect I believe there are six figures in total, not five

In the Discussion, I would suggest moderating statements such as ‘RPE orchestrates a chemotactic axis’, since the current evidence is based on *in vitro* assays and may not fully capture the *in vivo* complexity.

Similarly, the statement that ‘systemic antigen encounter conditions the ocular microenvironment’ should be nuanced, as only isolated pRPE were studied.

In the Conclusion, I recommend using ‘our findings suggest’ and explicitly acknowledging the need for validation in organoid or animal models before clinical translation.

In section 3.3, the detection of IL-2 under non-migratory conditions should be explained in more detail.

It is also unclear why the values obtained for exposure to live tachyzoites and conditioned supernatant appear to be the same—could the authors clarify this?

For the co-culture experiments (section 3.4), please specify whether RAW cells infected alone were included as controls.

Regarding the RPE experiments (sections 3.5–3.6), the data are interesting, but please double-check the citation of Figure 4, as it seems incorrect.

I believe there are six figures in total, not five.

**Discussion**

I suggest moderating statements such as “RPE orchestrates a chemotactic axis,” since the evidence provided is based solely on *in vitro* assays and may not fully reflect the *in vivo* context.

Similarly, the claim that “systemic antigen encounter conditions the ocular microenvironment” should be nuanced, as only isolated pRPE were studied.

I recommend rephrasing to use wording such as “our findings suggest,” and explicitly acknowledging the need for validation in organoid or animal models before any clinical translation.

## AUTHORS’ RESPONSE TO THE REVIEWERS

“In the Materials and Methods section, the authors refer to the ARPE-19 cell line, but in the Introduction and other parts of the text they use the term ‘RPE cells.’ Please ensure consistency throughout the manuscript. If the authors intend to refer to primary RPE (pRPE) cells, this should be clarified.”

Cell types were clarified and consistently identified as either pRPE or ARPE-19 throughout the manuscript.

The introduction is clear for the paper, linking RPE and immunity with *T.gondii* pathogenesis.

It would be helpful to clarify if the RPE migration toward infected foci is based on *in vivo* or *in vitro*, or both (Line 42).

A paragraph was added clarifying that RPE migration was evaluated under *in vitro* conditions.

In several figures, the panel letters appear incomplete or difficult to read.

Please revise and replace the labels to ensure they are clear and consistent, which will improve figure interpretation.

All figures were revised, and panel labels were replaced to ensure clarity and consistency

The section is generally well described, but it is unclear why the RH strain was used for the antigen preparation while ME49 was used for culture infection.

In the co-culture assays, it would be helpful to provide details on cell densities and how cell viability was controlled.

In addition, I recommend clarifying the number of experimental replicates in the Statistical Analysis section.

A paragraph was added explaining the rationale for using RH and ME49 strains.

The number of experimental replicates was specified in the Statistical Analysis section.

**Results:**

The findings are relevant and interesting but poorly described.

The description of Figure 1 should specify where the cells migrated, under what conditions, and provide appropriate comparisons.

More details are needed. If the authors only state that ‘migration’ occurred, it could be erroneously interpreted that the cells migrated to the bottom.

Did the cells migrate to the bottom of the Transwell insert, or to another location?

This information should be incorporated into the main text rather than left only in the figure legend.

The paragraph describing Figure 1 was rewritten to include details on migration sites, conditions, and comparisons.

I recommend avoiding the expression ‘cell displacement,’ as it may lead to misinterpretation (for example, section 3.2 Chemotaxis … ).

Both tachyzoites and ARPE cells are cells, so the term is ambiguous.

Given that the results are not clearly described, this type of wording makes it particularly difficult to follow the interpretation.

Terminology was revised to avoid ambiguity and improve precision.

Some of the figures are missing figure numbers, which makes the revision process difficult.

For example, the figure titled ‘Migration of primary retinal pigment epithelium’ should be properly numbered to ensure consistency and clarity throughout the manuscript.

Figure 1 (Section 3.1): Please correct the letter ‘B,’ which currently looks like an ‘E.’ In addition, specify how pore occlusion was quantified and indicate the number of replicates performed.

Figure numbers were corrected and standardized throughout the manuscript.

The letter was corrected. Quantification methods and the number of replicates were included in the text.

Figure 2 (Section 3.1): Please also correct the figure panel letters to ensure clarity.

Figure 2 panel letters were corrected for consistency and clarity.

Figure 3 (Section 3.2, under-agar assay): If possible, I suggest repeating the control conditions for ARPE-19, ARPE-19S, ARPE-19 (antigen 0.5), and ARPE-19 (antigen 1.0).

I also recommend maintaining the same scale when comparing migration differences at 24 h vs. 48 h for consistency and clarity.”

Triplicate numbers and experimental conditions were indicated for each situation. The same scale was maintained across comparisons for consistency.

In section 3.3, the detection of IL-2 in non-migratory conditions should be explainded in more detail, and I don’t understand why the values of exposure to live tachyzoites and conditioned supernatant are the same, Can you clarify that?

A paragraph was added explaining IL-2 detection results under non-migratory conditions.

For the co-culture experiments (section 3.4), it would be helpful to specify whether RAW cells infected alone were included as controls.

Regarding the RPE experiments (sections 3.5–3.6), the data are interesting, but please double-check the citation of Figure 4, as it seems incorrect I believe there are six figures in total, not five

Figure numbering was verified and corrected throughout the manuscript.

In the Discussion, I would suggest moderating statements such as ‘RPE orchestrates a chemotactic axis’, since the current evidence is based on *in vitro* assays and may not fully capture the *in vivo* complexity.

Similarly, the statement that ‘systemic antigen encounter conditions the ocular microenvironment’ should be nuanced, as only isolated pRPE were studied.

Statements were revised to moderate the conclusions and acknowledge the *in vitro* limitations.

In the Conclusion, I recommend using ‘our findings suggest’ and explicitly acknowledging the need for validation in organoid or animal models before clinical translation.

The suggested phrasing was adopted, and a note on future validation in organoid and animal models was included.

---

## [Reviewer Report · REVIEWERS COMMENTS]

## REVIEWER #1

The authors have addressed my previous comments, including revising figures, correcting numbering, and adding data to improve the clarity of the manuscript description.

They have also added sentences to moderate their conclusions and acknowledge the limitations of the *in vitro* model.

I would only like to suggest three minor points:

1) In my previous comment, I mentioned:

“The description of Figure 1 should specify where the cells migrated, under what conditions, and provide appropriate comparisons. More details are needed. If the authors only state that ‘migration’ occurred, it could be erroneously interpreted that the cells migrated to the bottom. Did the cells migrate to the bottom of the Transwell insert or to another location? This information should be incorporated into the main text rather than left only in the figure legend.”

However, the authors did not present any corresponding results; they only described a result without showing supporting data.

Data not shown should either be presented in the main text or included as supplemental material.

2) It will also benefit the manuscript if the authors clarify why and how they decided to use a sensitized cell line.

3) Finally, please standardize the term “primary RPE” as “pRPE” throughout the text.

## AUTHORS’ RESPONSE TO THE REVIEWERS

Response to the Editor

As requested, a justification regarding the ARPE-19 cell line was added at the end of section 2.2.

Experimental infection of ARPE-19 cells.

The term Primary RPE was standardized as pRPE throughout the manuscript.

A new result describing the final destination of the experimental cells—previously referred to as data not shown —was included (Figure 2A–C) and is now explained in section 3.1.

ARPE-19 migration in Transwell assays. Accordingly, the legend of Figure 1 was revised, and Figure 2 was added.

All figures were renumbered, and the legend of Figure 4 was rewritten for clarity.

---

## [Reviewer Report · REVIEWERS COMMENTS]

## REVIEWER #1

The authors have adequately addressed all the aspects previously suggested.

The abstract is clear and accurately reflects the scope and findings of the work.

The contribution remains original and relevant, representing an important addition to the field.

The methodology, results, and discussion are coherent and well-structured, providing sufficient detail to support the conclusions.

Results in general are clearly presented, informative, and consistent with the text.